# Steerable Generative Modeling of Playing Style at Scale

## Abstract

There has been a growing interest in using AI to model human behavior, particularly in domains where humans interact with this technology. While most existing work models human behavior at an aggregate level, our goal is to model behavior at the individual level. Recent approaches to *behavioral stylometry*—or the task of identifying a person from their actions alone—have shown promise in domains like chess, but these approaches are either not scalable (e.g., fine-tune a separate model for each person) or not generative, in that they cannot actually generate any actions. We address these limitations by framing behavioral stylometry as a multi-task learning problem—where each *task* represents a distinct *person*—and use parameter-efficient fine-tuning (PEFT) methods to learn an explicit *style vector* for each person. Style vectors are generative: they selectively activate shared "skill" parameters to generate actions in the style of each person. They also induce a latent space that we can interpret and manipulate algorithmically. In particular, we develop a general technique for *style steering* that allows us to steer a player's style vector towards a desired property. We apply our approach to two very different games, at unprecedented scales: chess (47,864 players) and Rocket League (2,000 players).

## 1 Introduction

Contemporary machine learning systems hold the promise of making a wide array of technologies more accessible and useful. Many of these systems are far more capable than the average person on domains for which they were trained, but they can often still be made more useful by better understanding how humans approach these same tasks. Rather than focusing on maximizing raw performance, such an understanding can help identify areas for improvement in humans, develop better AI partners or teachers, and create more enjoyable experiences. AI that solely aims to solve a task optimally often fails in these respects, because they tend to be difficult to interpret, provide limited instructional value, and can be awkward to interact with.

A common method for capturing human behavior is behavioral cloning (BC), a form of imitation learning (Schaal, 1996) that applies supervised learning to fixed demonstrations for a given task. BC has been used in various domains, such as supply chains (Kurian et al., 2023), legal cases (Ma et al., 2021), robotics (Florence et al., 2022), and self-driving vehicles (Pomerleau, 1988). Recently, BC has seen increasing use in gaming, reaching impressive performance in games such as Counter-Strike (Pearce & Zhu, 2022), Overcooked (Carroll et al., 2019), Minecraft (Schäfer et al., 2023), Bleeding Edge (Pearce et al., 2024; Kanervisto et al., 2025), and chess McIlroy-Young et al. (2020).

These works focus on modeling human behavior in aggregate, motivated by goals like developing better AI partners or training tools. This paper argues that such goals are better served by modeling human behavior at the *individual* level, allowing us to tailor solutions to an individual's specific needs (e.g., creating an AI training partner that targets an individual's weaknesses). To that end, recent work in chess has shown the most promise. McIlroy-Young et al. (2020) used BC to create a set of models called Maia that mimic human play at nine skill levels. They then fine-tuned these models on data from 400 individual players to create a personalized model per player (McIlroy-Young et al., 2022b). Using these models, the authors perform *behavioral stylometry*, where the goal is to identify which person played a given query set of games: they apply each of the 400 models to the query set and select the one with the highest move-matching accuracy. In follow-up work, McIlroy-Young et al. (2021) propose a more scalable approach—training a Transformer-based embedding on the games of each player. They perform stylometry across 2,844 players by embedding the query set of games and matching it to the closest player's corresponding embedding.

Table 1: Comparison of different approaches for player modeling. We combine the best of previous methods while providing novel capabilities.

| Criterion | Maia-MHR (Ours) | Embedding (e.g., McIlroy-Young et al. (2021)) | Personalized Models (e.g., McIlroy-Young et al. (2022b)) |
|---|---|---|---|
| Generative | $\checkmark$ | $\times$ | $\checkmark$ |
| Efficient Training | $\checkmark$ | $\checkmark$ | $\times$ |
| Efficient Stylometry | $\checkmark$ | $\checkmark$ | $\times$ |
| Player Synthesis | $\checkmark$ | $\times$ | $\times$ |
| Player Steering | $\checkmark$ | $\times$ | $\times$ |

These methods have varying merits, as summarized in Table 1. The individualized approach creates a generative model for each player that can play in their style, but it is not scalable—adding a new player requires fine-tuning a separate model—and it provides only a crude notion of stylometry (move matching). The embedding approach scales much better—it learns a single-vector representation of each player, and uses few-shot learning to embed new players in this space—but it cannot generate actions and hence cannot reason about behavior in practice.

An ideal solution would combine these properties to obtain a generative and scalable representation. Our key insight for achieving this is to view behavioral stylometry as a multi-task learning problem, where each *task* represents an individual *person*. The goal is to generalize across an initial set of players while supporting few-shot learning of new players. To do this efficiently, we leverage recent advances in parameter-efficient fine-tuning (PEFT) (Ponti et al., 2023; Caccia et al., 2023). Specifically, we augment an existing BC model with a set of shared Low Rank Adapters (LoRAs) and a routing matrix that specifies a distribution over these adapters for each player. Unlike approaches that train a separate LoRA for each task, this modular design allows players to softly share parameters in a fine-grained manner. We apply this adapter framework to two very different gaming models: a modified version of the Maia model for chess, and a Transformer-based model for Rocket League, a 3D soccer video game played by cars in a caged arena. We chose these games because they have a large, public collection of human games that span a diversity of skill levels and styles.

We start by creating a base BC model for each of these domains (which, incidentally, outperforms the state-of-the-art BC models in each domain). We then apply our adapter framework to the frozen model and fine-tune it: this encourages the adapters to learn different *latent skills*, while each row of the routing matrix learns a weight distribution over these skills. We call these rows *style vectors*, as they capture the underlying characteristics of each player's behavior. Style vectors are versatile and powerful. They support few-shot learning which enables stylometry at scale. They induce a generative model for each player that we can run and observe. Importantly, because style vectors are compact and generative, we can interpret and manipulate them algorithmically. We leverage these properties to develop a general, human-interpretable technique for *style steering* that identifies players who exhibit a desired style property, and steers a new player towards—or away from—that property.

In summary, we make the following contributions:

1. We develop an adapter framework to model individual human behavior and create a style vector for each player. We show that style vectors can be combined, interpolated, and steered in an interpretable way.
2. We apply our adapter framework and conduct behavioral stylometry at an unprecedented scale—47,864 players in chess and 2,000 in Rocket League. Our stylometry framework enables gradient-based search in the latent space, yielding orders-of-magnitude efficiency gains over prior work. Notably, it requires no stylometry-specific losses, allowing us to preserve full move-generation capabilities.
3. We present novel capabilities of style vectors, including a method to steer player styles to strengthen (human-interpretable) attributes of their gameplay. We show the generality of style steering by applying it to a very different domain: image generation for 10,177 celebrities.

## 2 BACKGROUND AND FRAMING

We frame behavioral stylometry and per-player generative modeling as a multitask learning problem. In multitask learning (Caruana, 1997; Ruder et al., 2019), we are given a collection of tasks $\mathcal{T} = (\mathcal{T}_1, \ldots, \mathcal{T}_{|\mathcal{T}|})$, where each task $\mathcal{T}_i$ is associated with a dataset $\mathcal{D}_i = \{(x_1, y_1), ..., (x_{n_i}, y_{n_i})\}$.

Multitask learning exploits the similarities among related tasks by transferring knowledge among them; ideally, this builds representations that are easily adaptable to new tasks.

The premise of this paper is that modeling individual human behavior from a pool of players can be interpreted as a multitask learning problem. Specifically, each task $\mathcal{T}_i$ consists of modeling the behavior of a specific player $i$. A dataset $\mathcal{D}_i$ corresponds to the sequence of game actions taken by player $i$, where each $(x, y)$ tuple denotes a game state $x$ and the action $y$ that player $i$ took in this state. We use the notion of *tasks* and *players* interchangeably.

## 2.1 PARAMETER-EFFICIENT FINE-TUNING

Popularized in NLP, parameter-efficient fine-tuning (PEFT) (Houlsby et al., 2019; Hu et al., 2022; Liu et al., 2022) has emerged as a scalable solution for adapting Large Language Models to several downstream tasks. Indeed, standard finetuning of pretrained LLMs requires updating (and storing) possibly billions of parameters for each task. PEFT methods instead freeze the pretrained model and inject a small set of trainable task-specific weights, or "adapters."

One such approach is the use of Low Rank Adapters (LoRA) (Hu et al., 2022), which modify linear transformations in the network by adding a learnable low-rank shift

$$h = \left(\boldsymbol{W}_0 + \Delta\boldsymbol{W}\right) x = \left(\boldsymbol{W}_0 + \boldsymbol{A}\boldsymbol{B}^T\right) x. \tag{1}$$

Here, $\boldsymbol{W}_0 \in \mathbb{R}^{d \times d}$ are the (frozen) weights of the pre-trained model, and $\boldsymbol{A}, \boldsymbol{B} \in \mathbb{R}^{d \times r}$ the learnable low-rank parameters of rank $r \ll d$. With this approach, practitioners can trade-off parameter efficiency with expressivity by increasing the rank $r$ of the transformation.

## 2.2 POLYTROPON AND MULTI-HEAD ADAPTER ROUTING

Standard PEFT methods such as LoRA can adapt a pretrained model for a given task. In multitask settings, training a separate set of adapters for each task is suboptimal, as it does not enable any sharing of information, or *transfer*, across similar tasks. On the other hand, using the same set of adapters for all tasks risks *negative interference* (Wang et al., 2021) across dissimilar tasks. Polytropon (Ponti et al., 2019) (`Poly`) addresses this transfer/interference tradeoff by softly sharing parameters across tasks. That is, each `Poly` layer contains 1) an inventory of LoRA adapters

$$\mathcal{M} = \{\boldsymbol{A}^{(1)}\boldsymbol{B}^{(1)}, \, \dots \, , \, \boldsymbol{A}^{(m)}\boldsymbol{B}^{(m)}\}$$

with $m \ll |\mathcal{T}|$, and 2) a task-routing matrix $\boldsymbol{Z} \in \mathbb{R}^{|\mathcal{T}| \times m}$, where $\boldsymbol{Z}_\tau \in \mathbb{R}^m$ specifies task $\tau$'s distribution over the shared modules. This formulation allows similar tasks to share adapters, while allowing dissimilar tasks to have non-overlapping parameters. The collection of adapters $\mathcal{M}$ can be interpreted as capturing different facets of knowledge, or *latent skills*, of the full multitask distribution.

At each forward pass, `Poly` LoRA adapters for task $\tau$ are constructed as

$$\boldsymbol{A}^\tau = \sum_i \alpha_i \boldsymbol{A}^{(i)}; \; \boldsymbol{B}^\tau = \sum_i \alpha_i \boldsymbol{B}^{(i)}, \tag{Poly}$$

where $\alpha_i = \texttt{softmax}(\boldsymbol{Z}\,[\tau])_i$ denotes the mixing weight of the $i$-th adapter in the inventory, and $\boldsymbol{A}^{(i)}, \boldsymbol{B}^{(i)}, \boldsymbol{A}^\tau, \boldsymbol{B}^\tau \in \mathbb{R}^{d \times r}$. Here, the $\tau$-th row of the routing matrix $\boldsymbol{Z}$ is effectively selecting which adapter modules to include in the linear combination. In our setting, where each task consists of modeling an individual, $\boldsymbol{Z}\,[\tau]$ specifies which latent skills are activated for user $\tau$; we call this their *style vector*. As per Eqn 1, the final output of the linear mapping modified with a `Poly` LoRA adapter becomes $h = \left(\boldsymbol{W}_0 + \boldsymbol{A}^\tau(\boldsymbol{B}^\tau)^T\right) x$.

In `Poly`, the module combination step is *coarse* as only linear combinations of the existing modules can be generated. Caccia et al. (2023) propose a more fine-grained module combination approach, called Multi-Head Routing (`MHR`), which is what we use in our work. Similar to Multi-Head Attention (Vaswani et al., 2017), the input dimension of $\boldsymbol{A}$ (and output dimensions of $\boldsymbol{B}$) are partitioned into $h$ heads, where a `Poly`-style procedure occurs for each head. The resulting parameters from each head are then concatenated to recover the full input/output dimensions. See A.2.

**Routing-only fine-tuning.** While LoRA adapters can reduce parameter costs from billions to millions, training the adapters for each new task can still be prohibitive when dealing with thousands of tasks. To this end, Caccia et al. (2023) proposed routing-only fine-tuning, where after an initial phase of pre-training, the adapter modules are frozen, and only the routing parameters $\boldsymbol{Z}$ are learned for a new task. We use this method for few-shot learning.

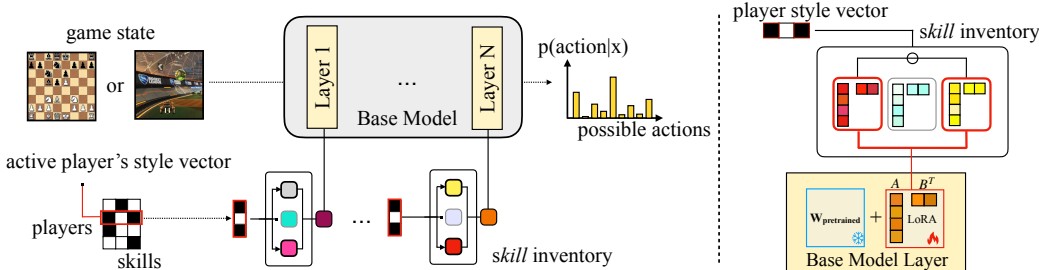

Figure 1: (**left**) Our overall architecture. We augment a base model with a set of MHR adapters and a routing matrix composed of each player's style vector. (**right**) Detailed view of an MHR layer, showing a skill inventory of adapters shared across players. The player's style vector specifies which skills are active (in this case, the first and third) to generate the final low-rank weight shift that is applied to the (frozen) base model layer.

## 3 ML METHODOLOGY

We detail our methodology for creating a generative model of individual behavior that enables our style analyses. We start with a base model and apply the `MHR` adapter framework to it, and then discuss model training and evaluation.

### 3.1 MODEL ARCHITECTURE

For chess, we follow McIlroy-Young et al. (2022b) and use the Squeeze-and-Excitation (S&E) Residual Network (Hu et al., 2018) as a base model, but with a deeper and wider configuration (see A.6). The total parameter count for chess is 15.7M. For Rocket League, we use the GPT-2 architecture from Radford et al. (2019), using a linear projection layer from the game state to the hidden dimensionality in lieu of any embedding layers. The total parameter count for Rocket League is 87.7M. For architectural specifics (e.g., attention heads, layer counts, hidden dimensions), see A.8.

To enable user-based adaptation, we incorporate the `MHR` adapters described in §2.2 into our base models, as shown in Fig. 1. For chess, we attach `MHR` adapters to each linear transformation used for channel-wise rescaling, for an additional 5M parameters. For Rocket League, we attach adapters to the fully connected layers of each transformer block, resulting in an additional 13.8M parameters. For all `MHR` adapter layers within a model, we share the same routing style vector. For specific details on adapter hyperparameters, see A.3.

### 3.2 DATA COLLECTION AND PARTITIONING

We use data from the largest open-source online chess platform, Lichess.org (Duplessis, 2021), which boasts a database of over 4.8 billion games. We collected Blitz games played between 2013 and 2020 inclusive—these are games with 3 or 5 minutes per side, optionally with a few seconds of time increment per move—and applied the same player filtering criteria as McIlroy-Young et al. (2022b). The resulting dataset comprises 47,864 unique players and over 244 million games. (See A.6 for a discussion on data imbalance.) For Rocket League, we collect data from a large open-source replay database, Ballchasing.com (CantFlyRL, 2024). We use 2.2 million 1v1 replays from 2015 to mid-2022, totalling several decades of human gameplay-hours. A.8 contains detailed information about the state and action space of the Rocket League game environment and data, along with the processing required to utilize the data.

We divide the set of players into a few subsets to support our training methodology. The *base player* set comprises all data and is used to train the base models. The *fine-tuning player* set is used to fine-tune the `MHR` architecture shown in Fig. 1. (For both, we split each player's data into 80/10/10 for train/test/validation.) The *few-shot player* set is used for few-shot learning based on a reference set of 100 games per player. For our chess experiments, to enable a direct comparison with prior work, we create an additional fine-tuning player set consisting of the same 400 players from those studies. Currently, we treat each player's data holistically, but in principle one could partition a player in different ways to analyze their playing style (see A.9).

### 3.3 MODEL TRAINING AND EVALUATION

**Base model pretraining.** We pre-train our base Maia model for chess using data from a base player set of all 47,864 players. Here, no player conditioning is performed during training, and no adapters are used. We treat this as a classification task of predicting human move $y$ made in chess position $x$,

given a datapoint $(x, y)$. We use the same loss functions and evaluation criteria as the original Maia work: Maia's policy head uses a cross entropy loss while the value head uses MSE; the output of the policy head is used to evaluate the model's move-matching accuracy.

We train our Rocket League model using a base player set of over 800,000 players, though the vast majority of players have 5 games or fewer. We discretize the actions into 3 bins for throttle, steer, pitch, yaw, and roll, as most of this data is close to 0, -1, or 1. We use binary outputs for jump, boost, and handbrake. A next-move prediction is labelled correct if and only if all the outputs are correct.

**MHR fine-tuning.** To train the MHR LoRA adapters, we adopt the methodology used in Caccia et al. (2023): namely, we freeze the base model and fine-tune the MHR layers and routing matrix using data from a fine-tuning player set. Recall that the routing matrix $Z$ has a row (style vector) for each player in the fine-tuning set. Following Ponti et al. (2019), we use a two-speed learning rate, where the style vector learning rate is higher than the adapter learning rate.

For chess, we use two fine-tuning player sets in our experiments, creating two separate MHR-Maia models. The first set comprises all 47,864 players and is used to evaluate behavioral cloning and stylometry at very large scale. The second set is comprised of the same 400 players used by McIlroy-Young et al. (2022b), which we use to compare few-shot learning and stylometry results. For Rocket League, we train an MHR-Rocket model on a fine-tuning set of 2,000 players with 100 games each.

**Few-shot learning.** To perform few-shot learning on our MHR models, we perform the "routing-only fine-tuning" described in § 2.2 that freezes all MHR LoRA adapters. Given a new player, we add a (randomly-initialized) row to $Z$ and fine-tune it on the player's reference set of games, eventually learning a style vector for the player while keeping other learning machinery fixed. Using this representation, we can generate sequences in the style of the corresponding player on held-out games and evaluate move-matching accuracy, as described above. To perform stylometry, if the player is a *seen* player (i.e., part of the fine-tuning set), then a matching style vector already exists in $Z$, and we can find it using cosine similarity. Otherwise, if the player is *unseen*, then we simply repeat the few-shot learning process on a query set of games (from the same player), and compare this new style vector to the entries in $Z$. In general, the number of reference/query games required for few-shot learning is low (see Figure 10, A.6).

For chess, (unless stated otherwise), all of our few-shot experiments use the MHR-Maia model fine-tuned on the 400-player set from McIlroy-Young et al. (2022b). For Rocket League, the few-shot player set consists of 1,000 of the 2,000-player set used to fine-tune MHR-Rocket.

**Evaluation.** We evaluate a fine-tuned MHR model in two ways. First, we measure its move-matching accuracy, similar to how we evaluate the base models. However, since our MHR models provide a generative model for each player (conditioned on their style vector), we can separately evaluate each player's model by applying it to their test set. We then average these per-player accuracies to determine the overall move-matching accuracy for the model.

Our second evaluation method uses the model to perform behavioral stylometry among all players in the fine-tuning set. To do this, we leverage our few-shot learning methodology. That is, given a query set of games from some player, we learn a new style vector in $Z$ for those games via few-shot learning, and compare this vector to every other vector in $Z$ using cosine similarity. We then output the player with the highest cosine similarity. In domains that focus on authenticating individuals (e.g., biometrics), ROC curves and related metrics are used. Our results can be re-interpreted in this way (see Figure 12 in the appendix).

## 4 STYLE METHODOLOGY

The style vectors in $Z$ give us a starting point for comparing player styles. For example, our stylometry method above uses the cosine similarity between vectors to determine how similar or different players are.

Style vectors can also be learned for different partitions of a player's dataset, or even for a merged dataset comprising multiple players. The latter is notable because it actually creates a new (human-like) playing style that has never been seen before. This suggests a general approach to synthesizing new styles: interpolate between existing players using a convex combination of their style vectors. To determine the playing strength of a newly synthesized player, we can simulate games between them and the players they are derived from, by conditioning the MHR model on their respective style vectors. The results of these games yield a win rate, which can be converted to a strength rating.

Table 2: Top-1 stylometry accuracy results. Query players are the game sets sampled from anonymous players that we would like to identify from the pool of all known candidate players. Game count is the number of games used to identify each query player. Random (%) indicates the chance of choosing the correct player when randomly sampling, and defines how difficult the task is. Numbers for McIlroy-Young et al. (2022b) and McIlroy-Young et al. (2021) are borrowed from their respective papers; the same 400 player dataset is used across all comparisons.

| Method | Query players | Candidates | Games | Random (%) | Acc. (%) |
|---|---|---|---|---|---|
| *Prior work comparison* | | | | | |
| McIlroy-Young et al. (2022b) | 400 | 400 | 100 | 0.25 | 98.0 |
| McIlroy-Young et al. (2021) | 400 | 400 | 100 | 0.25 | 99.5 |
| MHR-Maia | 400 | 400 | 100 | 0.25 | **99.8** |
| McIlroy-Young et al. (2022b) | 400 | 400 | 30 | 0.25 | 94.0 |
| MHR-Maia | 400 | 400 | 30 | 0.25 | **98.8** |
| *Scaling experiments* | | | | | |
| MHR-Maia (seen) | 10000 | 47864 | 100 | 0.002 | 94.4 |
| MHR-Maia (unseen few-shot) | 10000 | 10000 | 100 | 0.01 | 87.6 |
| McIlroy-Young et al. (2021) (unseen few-shot) | 578 | 2844 | 100 | 0.035 | 79.1 |

Currently, our advanced style synthesis techniques focus on chess, where simulating games is cheap and evaluation heuristics are standardized. Rocket League simulations are too costly at present for this, and there are no standardized heuristics, but in principle the same methodology can be applied.

In order to make style comparisons more interpretable, we draw inspiration from the concept probing technique used to analyze AlphaZero (a deep reinforcement learning chess engine) (McGrath et al., 2022). We use a set of human-coded heuristic functions found in Stockfish (a traditional chess engine) to evaluate a player's model. These functions capture concepts such as: king danger, bishop pair utilization, material imbalance, and so on. By invoking a player's model on a fixed set of chess positions, we can measure the change in the heuristic functions before and after their chosen move, and use this to summarize how much emphasis the player places on the corresponding concept.

Combining the above methods, we propose a simple but general method for *steering* a player's style towards a specific, human-interpretable attribute $a$ (e.g., king danger), while limiting the changes to other attributes (so as to preserve their style). We summarize this method in Algorithm 1. We first collect a set of players $X$ who exhibit high values for attribute $a$—determined, for example, by running their generative models on a fixed set of game states. We extract the common direction among these players, by averaging their style vectors and subtracting the population average. This yields a *style delta vector* that can be added to any player's style vector to elicit the desired change.

## 5 EXPERIMENTS

We show that MHR-Maia matches prior methods in chess stylometry and behavior cloning at scale, extends to Rocket League for both stylometry and move prediction, enables analysis and control of player styles, which generalizes beyond gaming to personalized, steerable image generation.

### 5.1 BEHAVIORAL STYLOMETRY

We evaluate MHR-Maia on behavioral stylometry, where the goal is to identify which player (from a set of known *candidate* players) produced a given set of games, we refer to this as a *query player* or query dataset. We compare against two prior methods: (i) individual model fine-tuning (McIlroy-Young et al., 2022b), which trains a separate Maia model for each player and predicts by selecting the model with highest move-matching accuracy on the query set, and (ii) a Transformer-based embedding method (McIlroy-Young et al., 2021), which embeds players into a 512-dimensional style space and matches query embeddings to candidate players. The former is computationally prohibitive, requiring one model per player and inference on each query set; in Rocket League, this would imply tens of trillions of tokens. The latter is purely discriminative, unable to create new styles or play the game.

Our hybrid approach combines the strengths of both. We fit a new style vector on the query games using efficient MHR adapters (§ 5.2), then use this vector like an embedding for scalable style search via cosine similarity. For *seen* chess players, we sample 10,000 players and fit vectors from 100 of

their games. For *unseen few-shot* players, we train MHR-Maia on the 400-player dataset of McIlroy-Young et al. (2022b), sample 10,000 held-out players, fit vectors from 100 of their games, and apply the same procedure to their queries. As shown in Table 2, MHR-Maia achieves 87.6% accuracy on a large set of unseen players, outperforming McIlroy-Young et al. (2021) (79.1%) despite their smaller candidate pool of 2,844 players. Additionally, when increasing the candidate player set from 400 to 47,864 players, MHR-Maia's accuracy drops by only 5.4%.

We extend this evaluation to Rocket League, marking (to our knowledge) the first stylometry study in this domain. Using our few-shot methodology, we compute style vectors for 1,000 query players drawn from 5-minute 1v1 matches, and identify each against a 2,000-player candidate pool, with 100 training games each. MHR-Rocket attains **86.7%** accuracy, demonstrating robustness even in a complex environment.

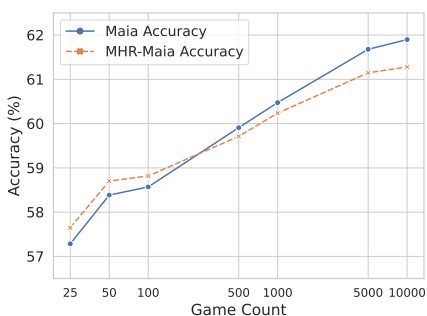

In contrast, training individual models per player proved infeasible: 1,000 players would require 2 million player comparisons, or inference over tens of trillions of tokens. An initial trial with 20 players and 100 games per player yielded 0% stylometry accuracy; scaling to 1,000 games per player (model training and query set) improved accuracy only to 50%—far below MHR-based style vectors. These results suggest that while individual models can succeed in structured domains like chess, they struggle

Figure 2: Move matching accuracy at various game counts of the individual models (Maia) and our method (MHR-Maia). MHR-Maia is within 1% accuracy of individual model fine-tuning using 1% of the compute cost.

in complex environments such as Rocket League, where the vast state space produces high variance and frequent out-of-distribution samples. By training a single shared model, MHR-Rocket is able to exploit the full dataset without splitting it into disjoint per-player subsets, enabling efficient use of training data to learn each player's style.

## 5.2 MOVE GENERATION

A key feature of our MHR models is that they are generative. We compare the efficacy of our method to using individually fine-tuned models for each player. We can not compare to the Transformer-based embedding method because it is incapable of generating moves. Full fine-tuning generally results in slightly improved performance compared to PEFT methods, albeit at much higher cost.

Nevertheless, Figure 2 shows that MHR-Maia performs comparatively well, achieving within 1% accuracy of individual model fine-tuning over a wide range of game counts. We achieve this using roughly 1% of the compute cost per player, as follows. On an A100 80GB GPU, training individual models required roughly 20 A100-minutes per player on average; thus, training on the full 47,864 player dataset would require thousands of A100-hours. In contrast, training MHR-Maia on the full dataset required roughly 7 A100-days, or around 12-13 A100-seconds per player, an improvement of nearly two orders of magnitude. Similar to Caccia et al. (2023), we found the training of style vectors to be relatively robust to the choice of hyperparameters; we discuss this in more detail in A.11.

The inference costs were roughly equal, with MHR-Maia being marginally more expensive due to the added parameters. The 47,864 player MHR-Maia model achieves a move prediction accuracy of 59.0%, while our base model achieves 54.4%. We further stress-test our method's generative capabilities and stylometry performance in A.7, where we consider players of the same skill level and show that we can still distinguish between their styles accurately.

For Rocket League, we compare the next move prediction of our base model (trained on over 800,000 players) with MHR-Rocket (fine-tuned on 2,000 players), to show that player-based conditioning via style vectors generates better predictions. We find that MHR-Rocket increases the next move prediction accuracy from **53.4%** to **56.6%**, matching the performance of individual model fine-tuning (**56.6%**, random performance being 0.05%).

## 5.3 ANALYSIS OF STYLE VECTORS

We explore the consistency of our style vectors within a player's data. We also compare playing styles using interpretable metrics, and generate new styles by averaging existing style vectors.

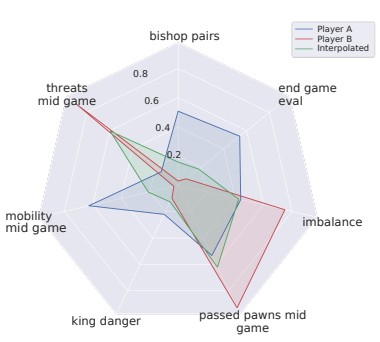

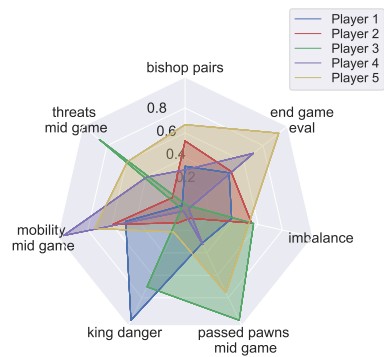

Figure 3: The style of an interpolated player (green) shown with the two component players (blue and red).

Figure 4: Comparing different player styles using human-interpretable evaluation metrics.

**Consistency within a player's data.** When a player's dataset is split into disjoint subsets, the resulting style vectors remain highly similar (Figure 7), showing that MHR captures player-specific traits regardless of how the data is partitioned. It should be possible to train multiple vectors per player to capture different game settings or sections per game, as we find that MHR-Maia and MHR-Rocket are not particularly sensitive to partitioning. To assess diversity, we sampled 5 random players, evaluated their move choices in $2^{17}$ positions, and report averaged Stockfish metrics in Figure 4. We provide further details on these experiments in A.5.

**Merged players.** When merging two players' datasets and retraining a style vector for them, we show that the resulting vector closely matches a simple average of the original players' vectors (Figure 8). This shows that style vectors can be combined to create intermediate player styles without retraining. As a case study, we average the style vectors of two randomly sampled players and evaluate 4096 games using Stockfish heuristics: we find that the new player (green) lies between the component styles (red, blue) of the original players (Figure 3).

### 5.4 SYNTHESIS OF NEW STYLES

We investigate applications of style synthesis that can help humans improve: interpolating weaker player styles to stronger ones, and steering player styles along human-interpretable properties.

**Interpolating between players.** We show that interpolating between the style vectors of a weaker and stronger player results in new players whose skill levels also interpolates between the players. Here, we take 100 pairs of weak and strong player style vectors and gradually interpolate between them as $(1 - \lambda)u_w + \lambda u_s, 0 \leq \lambda \leq 1$, where $u_w$ and $u_s$ are the respective vectors. For each value of $\lambda$, we simulate 1,000 games between the interpolated player and $u_s$, the stronger player. Figure 5 plots the win rate of the interpolated players as a function of $\lambda$ for each pair of players. This plot shows that the win rate increases in a roughly linear fashion as lambda increases, starting low and eventually winning roughly half the time, which is what we would expect from two players with the same style vectors. This allows us to create a continuous range of skill levels, unlike current models such as McIlroy-Young et al. (2020).

**Steering player style.** We can directly control the playing style of a player using the steering method described in §4. Using the human-interpretable Stockfish heuristics, we identify players in our chess dataset with high ($> 2$ std) bishop pair utilization, and similarly players with high king danger. We use these player sets to compute style delta vectors corresponding to these attributes, and then simply add the delta vectors to 2,000 randomly sampled players' existing style vectors. Figure 6 shows the change (normalized by the standard deviation for that attribute) in these players' Stockfish evaluations after adding the style delta vectors. We see that the player's style is steered towards the attribute in question, with modest impact on other attributes.

To examine whether our steering approach generalizes beyond gaming, we use our MHR framework to fine-tune Stable Diffusion 1.5 (Rombach et al., 2022) on the CelebA dataset (Liu et al., 2015), and apply the same delta-vector procedure to steer (edit) the generated images. We compare our approach to several popular image editing methods and show that it performs quite favorably (see A.4).

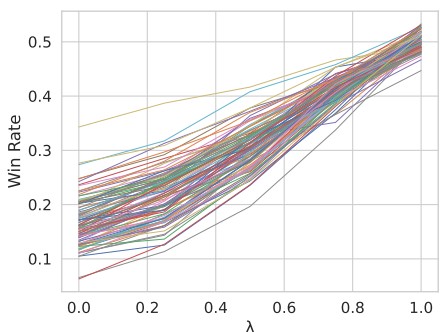

Figure 5: Win rate as randomly chosen weaker players are interpolated towards randomly chosen stronger players.

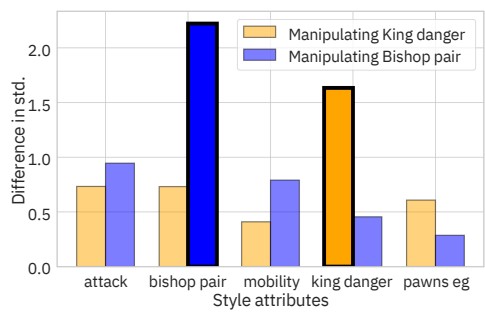

Figure 6: Using our style steering method to increase two Stockfish attributes (separately) for 2,000 random players.

## 6  OTHER RELATED WORK

**Stylometry and player style modeling.**    Originally used for author attribution via statistical analysis of text (Tweedie et al., 1996; Neal et al., 2017), stylometry has found broad application in tasks like handwriting recognition (Bromley et al., 1993), speaker verification (Wan et al., 2018), identifying programmers from code (Caliskan-Islam et al., 2015), determining age/gender from blog posts (Goswami et al., 2009), and identifying authors of scientific articles (Bergsma et al., 2012). In the context of gaming (covered in the introduction), stylometry is closely related to playstyle modeling, where the goal is to associate a player with a reference style, such as by building agents with different playstyles and finding the closest behavioral match (Holmgård et al., 2014), or gathering gameplay data and applying methods like clustering (Ingram et al., 2022), LDA (Gow et al., 2012), Bayesian approaches (Normoyle & Jensen, 2015), and sequential models (Valls-Vargas et al., 2015) to group players with similar styles. Kanervisto et al. (2021) characterizes an agent's behavior by analyzing the states that an agent sees (not actions). Khandelwal et al. (2024) fine-tunes a language model to predict verbalized user behaviors. Unlike our work, these approaches either focus on aggregate playstyles, or do not learn generative models of behavior that can be conditioned on an individual.

Our method for style synthesis is inspired by earlier work on vector arithmetic with embeddings (Church, 2017), and recent work on steering multitask models with task vectors (Ilharco et al., 2023). Our steering method is reminiscent of Radford et al. (2016), which manipulates the model's latent space to generate images containing specific attributes. Recently, Dravid et al. (2024) achieved similar results on CelebA by training LoRAs and manipulating their weights.

**Parameter-efficient adaptation.**    Adapter based methods inject (and update) new parameters within a pretrained model while keeping the backbone fixed. Houlsby et al. (2019) defines an adapter as a two-layer feed-forward neural network with a bottleneck representation. Similar approaches have been used for cross-lingual transfer (Pfeiffer et al., 2020). Adapters have also been used in vision based multitask settings (Rebuffi et al., 2017). More recently, Ansell et al. (2022) propose to learn sparse masks and compose them to enable zero-shot transfer. Hu et al. (2022) learn low-rank shifts on the original weights, and (Liu et al., 2022) learns an elementwise multiplier of the pretrained model's activations. Adapters have also been used in multitask settings. Chronopoulou et al. (2023) independently trains adapters for each task, and merges parameters of relevant tasks to transfer to new ones. Soft prompts (Lester et al., 2021) append learnable tokens to natural language sequences. Vu et al. (2021) learns a collection of soft prompts for multitask training sets that can be applied to new tasks.

## 7  CONCLUSION

We show that individual player behavior can be modeled at very large scale in games as different as chess and Rocket League. We cast this problem in the framework of multi-task learning and employ modular PEFT methods to learn a shared set of skills across players, modulated by a distinct style vector. We use style vectors to perform stylometry, analyze player styles, and synthesize and steer new styles. Our style methodology shows promise in domains outside of gaming, such as in image editing, which we plan to explore further in future work.

## 8 ETHICS STATEMENT

All players used in our submission have been anonymized and no personally identifying information is shown. Our data is collected from sources that make it clear to players that their data will be made public for use by the community. Prior work has analyzed the risks and benefits of models trained to mimic individual style in McIlroy-Young et al. (2022a).

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

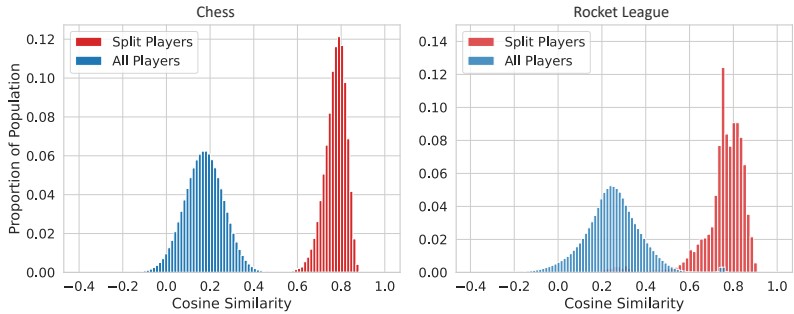

Figure 7: Cosine similarity between style vectors from different partitions of the same player (red) versus across players (blue). Vectors from a single player are markedly more similar, confirming that player styles are unique and identifiable regardless of data partitioning.

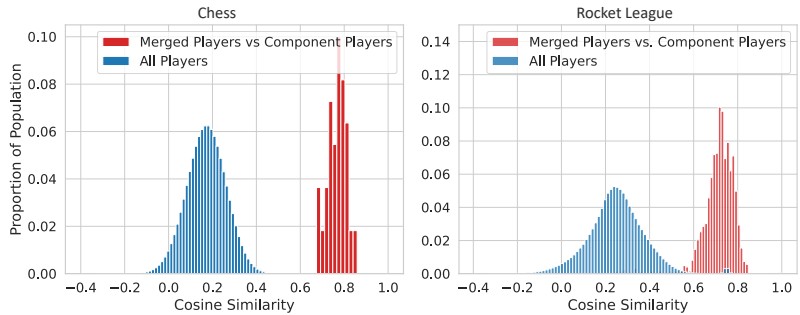

Figure 8: Cosine similarity between averaged style vectors of two players and vectors learned on their merged datasets (red) versus across all players (blue), showing that style vectors can be combined to create intermediate player styles without retraining.

## A    APPENDIX

### A.1    ICLR LARGE LANGUAGE MODEL USAGE

Large language models (LLMs) were used to ensure grammatical correctness in some sections of the paper, and all outputs were thoroughly vetted and edited prior to being used.

### A.2    MULTI-HEAD ADAPTER ROUTING

In `Poly`, the module combination step remains *coarse*, as only linear combinations of the existing modules can be generated. Caccia et al. (2023) propose a more fine-grained module combination approach, referred to as Multi-Head Routing (`MHR`). Similar to Multi-Head Attention (Vaswani et al., 2017), the input dimension of $A$ (and output dimensions of $B$) are partitioned into $h$ heads, where a `Poly`-style procedure occurs for each head. The resulting parameters from each head are then concatenated, recovering the full input (and output) dimensions. This makes the module combination step *piecewise linear*, with a separate task-routing matrix $Z$ learned for each head.

Formally, a `MHR` layer learns a 3-dimensional task-routing tensor $\mathbf{Z} \in \mathbb{R}^{|\mathcal{T}| \times |\mathcal{M}| \times h}$. The 2D slice $\mathbf{Z}_{:,:,k} \in \mathbb{R}^{|\mathcal{T}| \times |\mathcal{M}|}$ of the tensor $\mathbf{Z}$ denotes the distribution over modules for the $k$-th head, and $\boldsymbol{W}[k] \in \mathbb{R}^{\frac{d}{h} \times r}$ the $k$-th partition along the rows of the matrix $\boldsymbol{W} \in \mathbb{R}^{d \times r}$. The adapter parameters $\boldsymbol{A}^\tau \in \mathbb{R}^{d \times r}$ for task $\tau$, and for each adapter layer, are computed as (similarly for $\boldsymbol{B}^\tau$):

$$\boldsymbol{A}_k^\tau = \sum_j \alpha_{i,k} \cdot \boldsymbol{A}_j[k] \ \text{ with } \ \boldsymbol{A}_k^\tau \in \mathbb{R}^{\frac{d}{h} \times r}, \tag{MHR}$$

$$\boldsymbol{A}^\tau = \texttt{concat}(\boldsymbol{A}_1^\tau, \dots, \boldsymbol{A}_h^\tau),$$

where $\alpha_{i,k} = \texttt{softmax}(\boldsymbol{Z}[\tau, :, k])_i$. Importantly, the number of LoRA adapter parameters does not increase with the number of heads. Only the task-routing parameters linearly increase with $h$ for `MHR` vs. `Poly`. However, this cost is negligible as the parameter count of the routing matrices is much smaller than for the LoRA modules themselves.

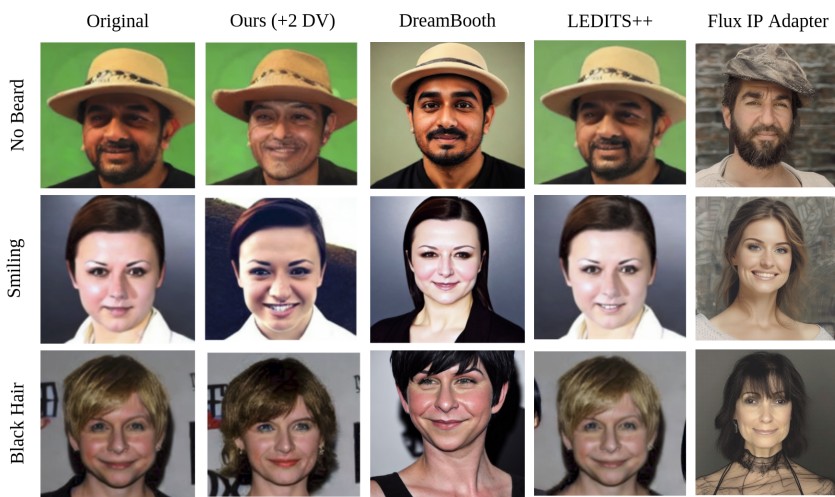

Figure 9: Images generated by steering Stable Diffusion 1.5 (Rombach et al., 2022) fine tuned with our method on the CelebA (Liu et al., 2015) dataset. We compare against using DreamBooth (Ruiz et al., 2023) on the original image and modifying the prompt, along with popular image editing methods LEDITS++ (Brack et al., 2024), and Flux-IP-Adapter (XLabs-AI, 2024).

### A.3 ADAPTER SETUP

In chess, for every linear transformation in the MLP used for channel-wise rescaling, we add an `MHR` layer built of LoRA adapters. with rank 16, for a total of $12 \times 2 = 24$ `MHR` layers. We use an adapter inventory of size 32 and a multi-head routing strategy with 8 heads. Therefore, for each user we must learn $32 \times 8 = 256$ routing parameters as their style vector; this yields 5M additional parameters. For Rocket League, we attach the adapters to the fully connected layer of each transformer block, resulting in 12 `MHR` layers of LoRAs with rank 16. We use an inventory size of 16 and 64 heads. This yields 13.8M additional parameters. To facilitate interpretability and style analysis, we use the same routing (style vector) across all `MHR` layers.

### A.4 STEERING DIFFUSION MODELS

To address questions about the generalizability of our method, we applied the exact style delta vector computation and steering algorithm outlined in § 4 to steer the outputs of an image generation diffusion model in a fine-grained manner. Specifically, we fine-tune style vectors for 10,177 identities from the CelebA Faces With Attributes dataset (Liu et al., 2015), using Stable Diffusion 1.5 (Rombach et al., 2022) as our base model. We then compute "No Beard," "Smiling," and "Black Hair" style delta vectors using cosine similarities between images and their corresponding CLIP embeddings (Radford et al., 2021). Figure 9 shows sample generations with and without steering, where the leftmost images are unaltered.

We compare against several strong baselines: DreamBooth (Ruiz et al., 2023) with LoRA using the scripts in Mangrulkar et al. (2022), Flux IP Adapter (XLabs-AI, 2024), and LEDITS++ (Brack et al., 2024). For DreamBooth, we fine-tune on the original image and modify the prompt (e.g., "with no beard," "smiling," "with black hair"). DreamBooth elicits the desired changes for "Smiling" and "Black Hair," but not "No Beard," and often alters unrelated aspects of the image. Flux IP Adapter and DreamBooth both change the style and subject significantly, reducing faithfulness to the source. LEDITS++ is able to elicit "Smiling," but despite trying extensive tuning of prompts and parameters, it fails for "No Beard" and "Black Hair."

In contrast, our method consistently produces all three edits while remaining faithful to the original image, achieving more granular control with minimal unintended changes. Importantly, these results are obtained without tailoring our algorithm to the image domain: we simply reuse the same delta-vector computation from chess stylometry. Despite the low fidelity of the CelebA images (128×128 resolution), our approach edits the images effectively, demonstrating both robustness and generality. We leave further application of this work in the image editing space for future work, as we believe that it would benefit from a more focused and in-depth analysis.

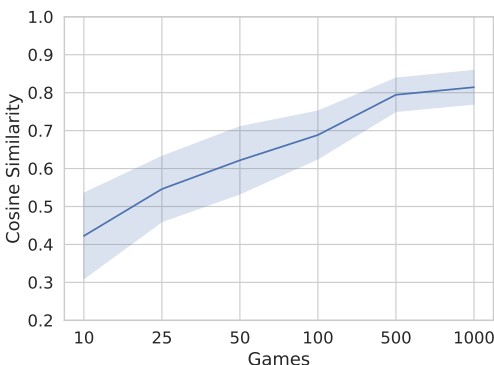

Figure 10: Cosine similarity of style vectors trained with varying game sizes compared to a style vector trained with 10,000 games, run on 50 players.

### A.5 EXPANDED STYLE VECTOR CONSISTENCY ANALYSIS

We first partition 50 players' datasets into disjoint subsets. We use 50 splits for chess and 20 for Rocket League. The subsets are sampled across a wide range of dates, opposing players, and playing sessions. We then train a style vector on every split, and compare these vectors using cosine similarity. We find that vectors corresponding to the same player are significantly more similar to each other than the general population, visualized in Figure 7. This suggests that our MHR models are able to associate distinct style characteristics with each player. These characteristics can be learned with relatively few games, as shown in Figure 10 and A.6.

### A.6 CHESS ARCHITECTURE/DATA

Our base Maia architecture follows McIlroy-Young et al. (2022b) and uses the Squeeze-and-Excitation (S&E) Residual Network of (Hu et al., 2018). At every residual block, channel information is aggregated across spatial dimensions via a global pooling operation. The resulting vector is then processed by a 2-layer MLP, with a bottleneck representation compressing the number of channels by $r$. The output of this MLP is a one-dimensional vector used to scale the output of the residual block along the channel dimension. We use 12 residual blocks containing 256 filters, and a bottleneck compression factor of $r = 8$. We note that this differs from the base Maia model in McIlroy-Young et al. (2022b), which uses 64 filters and 6 residual blocks. The input is a 112-channel $8 \times 8$ image representation of the chess board; the output is the predicted move encoded as a 1858-dimensional one-hot vector. The total parameters is 15.7M.

While our dataset has a median game count of 3,479 games, many players may have as few as 10-50 games, implying some degree of data imbalance. Our evaluation of few-shot learning shows that 100 games is sufficient to learn the style vector of an unseen player. However, one might still ask how accurately such a style vector is given a very small number of games. To explore this, we first split a player into disjoint sets of 10, 25, 50, 100, 500, and 1,000 games. We then train a style vector on each set. As a baseline, we train a style vector on 10,000 games and track the cosine similarity of the smaller-set style vectors relative to this baseline vector. We show the results in Figure 10. The single-tailed P-value for the similarity of a vector trained with just 10 games to one trained on all games for that player is roughly 0.02.

### A.7 SIMILAR ELO TESTING

To further test robustness, we evaluate stylometry performance among players with similar Elo ratings. We sample 100 players rated ~1300 and 100 players rated ~1700, and perform few-shot learning using MHR-Maia on 100 games each. Stylometry accuracy remains high even in this challenging setting: with 100 query games and 100 query players (4,000 total players per Elo bucket), the 1300 group achieves 100% top-1 identification accuracy and the 1700 group 99%. Move-matching accuracy also remains strong within Elo buckets (Table 3), with self-games consistently outperforming other-player games, reflecting the ability of our method to capture subtle stylistic signals even among players with similar skill level.

Table 3: Move-matching accuracy on players of similar Elo.

| Elo Bucket | Eval Group | Top-1 Acc (%) | Top-3 Acc (%) |
|---|---|---|---|
| 1300 | Self | 57.48 | 83.89 |
| 1300 | Others | 50.50 | 78.38 |
| 1700 | Self | 59.95 | 85.88 |
| 1700 | Others | 53.46 | 81.16 |

## A.8 ROCKET LEAGUE ARCHITECTURE/DATA

For Rocket League, we use the GPT-2 architecture from Radford et al. (2019) with a dimensionality of 768, 12 attention heads, and 12 layers. The input is a 49-dimensional vector with game physics information; the output is 8 heads: 5 with [-1, 0, 1] bins and 3 binary heads for a total of 1944 action combinations. The model has no embedding layer, as the game datapoints are passed directly as tokens after processing. The total parameters is 87.7M.

Our 1v1 replays dataset was scraped over the course of several weeks from the Ballchasing.com API using the Grand Champion subscription tier, though the API does have a slower free tier. This API yields raw game replays, which are uploaded by users either manually or using a community-made plugin for the game. The replays are in a binary format which must be parsed using community-made projects such as Carball (SaltieRL, 2024).

The Carball library allows us to convert the binary replay format to a more standard CSV format, which we save to a Cloud binary blob storage. The data present in both is a lossy reconstruction of game states, and requires some processing to be usable. In particular, the data is sampled at an inconsistent rate (varying between 24hz and 27hz), contains repeated physics ticks, and is missing action data for aerial controls (pitch, yaw, roll).

We resolve the issue of sampling rate and repeated ticks by removing repeated ticks, and doing a time-weighted resampling and interpolation to a standard 10hz for model training, though we found that 30hz also works well. Note that the actual game physics ticks occur at 120hz, so any value aligned with this should work. Without these changes, the model performs extremely poorly and is unable to navigate the arena.

We resolve the issue of missing aerial controls through the physics-based solver present in the Carball library. The estimation of these controls is not perfect, but it is sufficient for our purposes. Some previous community work has used inverse dynamics (Braaten, 2022) trained from rollouts of in-game bots to solve for these actions, though we opted to not use this due to the inconsistency in replay data sampling.

The data returned by the CSVs are fairly large, messy, and inconsistent. We apply the following transformations to the dataframe to bring the values closer to 0:

- Divide position by 2300

- Divide linear velocity by 23000

- Divide angular velocity by 5500

- Divide boost by 255

- Encode rotation Euler angles according to Zhou et al. (2020)

Additionally, when turning the data into tokens for use in our model, we add in an extra dimension to represent the team, and concatenate the opponent's data points along with the position, linear and angular velocity of the ball. We complete all of these transformations at runtime.

We also have to align the data returned by the simulators for Rocket League with the data used to train the model, RLBot (RLBot, 2017) and RLGym (Emery, 2021). Along with including an extra dimension to represent the team, we apply the following transformations to all samples obtained from the game:

- Divide position by 2300

- Divide linear velocity by 2300

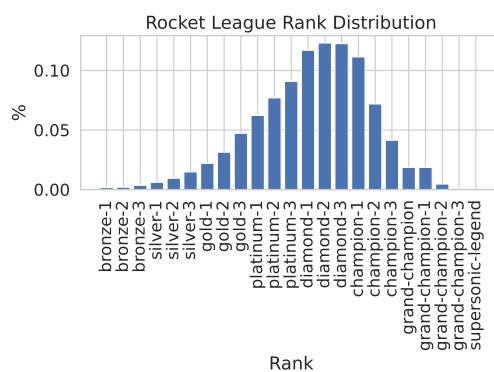

Figure 11: Skill distribution of Rocket League players in our dataset.

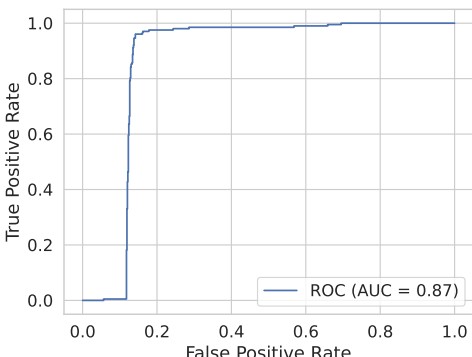

Figure 12: ROC Curve of Rocket League player detection. To generate this curve, any output other than the actual player is treated as an incorrect prediction.

- Divide angular velocity by 5.5

- Divide boost by 100

The skill distribution of the players in our dataset can be found in Figure 11. After parsing and processing, each Rocket League game state is a vector holding the player's 3D position, linear and angular velocity, boost remaining, rotation, and team; we also include the opponent's state and the position, linear and angular velocity of the ball. Given a game state, we must predict the user's throttle, steer (while grounded), pitch, yaw, roll (while aerial), jump, boost, and handbrake.

A.9 IMPLICIT STATIONARITY ASSUMPTIONS

Most prior work in chess stylometry assumes that a player's style is stationary over time and across gameplay situations. In practice, however, style may vary with the opponent, the chosen opening, or the stage of the game (opening, middlegame, endgame). For example, McIlroy-Young et al. (2021) show that removing the first 15 moves (the opening) reduces identification accuracy, suggesting that openings disproportionately influence style detection. Our approach does not rely on stationarity or handcrafted splits: in principle, one could partition a player's data into openings, middlegames, endgames, or even opponent defenses or time-of-day effects, and train distinct style vectors for each. Despite treating players holistically, we are able to capture the peculiarities of individual style and perform stylometry with high accuracy, directly comparable to prior holistic baselines.

Regardless, to test how strongly non-stationarity affects stylometry, we randomly sampled 20 players with at least two years of data. For each, we fine-tuned the base Maia model on the first 20% of their games, then evaluated move-matching accuracy on subsequent time windows (Figure 13). We observe a downward trend in accuracy as styles shift over time, but performance never falls below

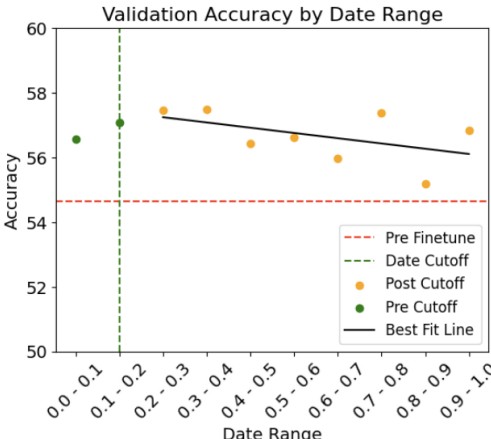

Figure 13: Accuracy of a model fine-tuned on data before a cutoff date, tested on much later dates. The efficacy of a personalized model is strong, even after years of continued playtime.

that of the base Maia model. This suggests that while certain elements of style evolve, others remain consistent enough to support strong prediction accuracy even in the presence of temporal drift.

## A.10 STYLE STEERING METHOD

---
**Algorithm 1** Style Delta Vector computation

---
**Input:**
  $X$ : Style vectors of top-k players for attrib. $a$;
  $P$ : Style vectors of all players in population
**Output** $\Delta_a$: Style delta vector for attr. $a$

  $V_a = \text{mean}(X, \texttt{axis} = \text{`players'})$
  $V_P = \text{mean}(P, \texttt{axis} = \text{`players'})$
  $\Delta_a = V_a - V_P$
  **Returns** $\Delta_a$

---

## A.11 HYPERPARAMETER SENSITIVITY FOR STYLE VECTOR TRAINING

We did a small search over a few of the hyperparameters for the architecture early on in the project, and found that the main deciders are: (1) LoRA size (MHR heads, rank), (2) using a shared routing matrix, and (3) learning rate for the routing matrix. The other hyperparameters should not be a significant concern as long as they are chosen reasonably. Increasing the number of skills can help produce more granular feature decompositions, but we find that the other parameters are more important for performance. We use a rank of 16 as that was the smallest adapter that did not result in significant underfitting. Using a shared routing matrix helps to create a more interpretable skill space, and increasing the routing matrix learning rate is crucial for training efficiency. We believe that our performance would improve by increasing the number of LoRA heads significantly, but we prefer to preserve computational cost for practicality reasons at these player scales.

