# OpenReview forum: "Steerable Generative Modeling of Playing Style at Scale"
_ICLR.cc/2026/Conference — ICLR 2026 Conference Withdrawn Submission_

### Official Review · Reviewer_NJqA · 2025-10-23

**Soundness:** 1
**Presentation:** 3
**Contribution:** 1
**Rating:** 2
**Confidence:** 4

**Summary:**

This paper proposes a mthod to learn individual player behavior by the multi-task setting--learning each individual separately. The authors incorporate the routing-only PEFT method to perform few-shot learning on new players and the experiments demonstrate that the proposed MHR-Maia learns player styles on Chess and Rocket.

**Strengths:**

1. The analyzing experiments in this paper are interesting and extensive, especially the qualitative trials to CelebA aiming to demonstrate generalizability.
2. The proposed method is reasonable and have potential to be applied to other domains.

**Weaknesses:**

1. It is appreciated that the authors examine the generalizability of their proposed method in Appendix A.4 and that Figure 9 looks comparative. However, it is insufficient to merely use 3 qualitative cases as cherry-picks (note that the authors explicitly mention *10177 celebrities in the third contribution) to make such claims convincing, especially that image editing has been well studied in recent years--I doubt that the proposed method could outperform or be on par with those generative methods with the corresponding benchmarks. If the authors' purpose is only to demonstrate generalizability, it is suggested to exclude other methods.
2. The experiments are only conducted with rocket, which may have less interests to the ICLR community. Note that there are other public sports data (e.g., badminton [1], basketball [2]) that can be used to validate generalizability and robustness of the proposed method. In addition, the authors are suggested to include quantitative experiments from other domains such as image editing.
3. The motivation of tackling player behaviors at the individual level is not convincing, since this setting requires much more data to train the model and cannot share common styles across players, which are common to be seen in sports (e.g., passive/active players). Additionally, using multi-task settings to learn each player cannot be generalized to unseen players. The paper argue for the usage (example in L45); however, a junior player who hasn't played many games cannot benefit from this method because the model cannot transfer common styles from other players to target this player's weaknesses.
4. Another intuitive method is to specify which player is playing together in the input sequence (e.g., [3, 4]), which does not require multi-task settings but preserving implicit information within the same model. The authors should clarify about the setting and compared with other literature developing player styles.
5. I didn't find novelty from this paper, which looks more like a development report. Although the authors depict three contributions, the first one has potential concerns to the motivation and the second contribution is built by using commonly used methods (e.g., routing-only finetuning).
6. The baselines are outdated, the newest one is from 2022. It would be more convincing if the authors could include more recent approaches (e.g., [4]).
7. The idea of measuring accurate style in Appendix A.6 is interesting. However, Figure 10 does not really measure if style vectors are *accurate* but *similar*. It is important to note that vectors similar to the ones trained with 10k games cannot be interpreted as accurate styles.
8. [Minor] The authors significantly adjust the margins in the paper to fit more contents in the main part, reducing readability. It would be better to move some contents to the appendix and provide proper reference.
9. [Minor] Using the term `unseen few-shot` is a bit awkward since few-shot learning aims to *see* references with a few samples.

[1] ShuttleSet: A Human-Annotated Stroke-Level Singles Dataset for Badminton Tactical Analysis

[2] PlayBest: Professional Basketball Player Behavior Synthesis via Planning with Diffusion

[3] ShuttleNet: Position-aware Fusion of Rally Progress and Player Styles for Stroke Forecasting in Badminton.

[4] Offline Imitation of Badminton Player Behavior via Experiential Contexts and Brownian Motion

**Questions:**

Q1: In L860-861, could the authors elaborate more on *reuse the same delta-vector computation from chess* and the claim about *without tailoring our algorithm to the image domain*? Did the authors apply the same algorithm but finetuning on the image domain? It's confusing because if it is the case, it still reflects the aforementioned concerns that this paper only evaluates on a specific dataset/domain.

Q2: For the few-shot player set (L208), are those players included in the base or finetuning player sets?

Q3: Since multi-task settings are difficult to stably train, how does the training loss behave?

---

> ### Author Response · Authors · 2025-12-03
>
> Thank you for your review. Regarding the concern about reusing the same algorithm across datasets, you are correct that we simply apply the exact same methodology for fine-tuning. However, the primary dataset requirement is that it can be split into meaningful, smaller subsets (e.g., per-player data), which we believe is relatively common. Additionally, evaluating a method on specific datasets or domains is standard in empirical research, and our goal is to show that the same approach works across distinct action-oriented environments. Rocket League and Chess differ substantially in structure, dynamics, and skill expression, so strong results in both indicate that the method is not tied to a narrow setting.
>
> For the few-shot player set, none of those players appear in the fine tuning data, ensuring that our evaluation genuinely measures generalization. Training was stable overall, although the loss required a relatively high learning rate to begin decreasing, suggesting a small hyperparameter search may be beneficial in practice.

---

### Official Review · Reviewer_Epd4 · 2025-10-29

**Soundness:** 3
**Presentation:** 2
**Contribution:** 2
**Rating:** 4
**Confidence:** 4

**Summary:**

This paper proposes a Low-Rank Adaptation (LoRA)-based parameter-efficient fine-tuning (PEFT) framework for modeling individual playing styles, building upon existing methods. For each player, the model learns a personalized style vector, which is used to activate corresponding Multi-Head Routing (MHR) adapters and generate player-specific behaviors. Experiments evaluated on Chess and Rocket League, achieving comparable accuracy to full fine-tuning while requiring only ~1% of the total compute - within 1% of Maia’s move-matching accuracy in Chess, and improving action prediction accuracy from 53.4% to 56.6% in Rocket League. Additionally, the learned style vectors are reported to enable player identification (behavioral stylometry), with recognition accuracy claimed to surpass prior methods. Overall, the paper primarily integrates existing techniques, including LoRA, MHR, and routing-only fine-tuning, into a new application domain, rather than introducing novel algorithmic contributions.

**Strengths:**

1. The paper is among the first to combine Parameter-Efficient Fine-Tuning (PEFT) and Multi-Head Routing (MHR) for style steering tasks.

2. In addition to standard experiments, the paper provides several analyses and applications of the learned style vectors.

3. The methodology is illustrated with a clear overall framework diagram.

4. The work demonstrates the feasibility of modeling multiple individual styles within a single shared model.

**Weaknesses:**

1. The critical weakness of this paper is its novelty, as it primarily combines existing techniques - Low-Rank Adaptation (LoRA), Multi-Head Routing (MHR), and routing-only fine-tuning - without introducing any new algorithmic or theoretical contributions.

2. The evaluation on Rocket League is limited in both scale and analysis, lacking detailed ablation studies or stylometry experiments to support the reported improvements.

3. The results are presented without error bars, variance, or statistical significance testing, making it difficult to assess whether the observed performance gains are reliable.

4. The stylometry comparison raises fairness concerns, since the proposed method performs few-shot fine-tuning for unseen players, whereas prior baselines operate in a zero-shot setting.

**Questions:**

1. Please address the concerns raised in Weaknesses.

2. In Table 2, the paper cites a 79.1% few-shot accuracy for McIlroy-Young et al. (2021). However, in the original paper, Table 2 reports 97.9% accuracy for unseen players at k = 0. Could the authors clarify whether 79.1% was obtained under a different setup or if this may be a reporting error?

3. Section 5.4 shows that combining a weak and a strong player's style vectors produces an intermediate strength. What would happen if two players of similar strength but distinct or even conflicting styles were combined? Would the resulting vector represent a neutral mixture, or could it exhibit non-linear or adversarial interactions between the two styles?

---

> ### Author Response · Authors · 2025-12-03
>
> Thank you for your review. Regarding novelty, we believe our work is the first to explore granular style control in action-oriented environments. Our method contributes a way to use population-level heuristics derived from game states and controls them in a precise and human-interpretable manner.
>
> On Rocket League stylometry, we agree that the analysis is not as extensive as our Chess study, but we do include stylometry experiments in Section 5.1, and they support the same findings observed in the more detailed chess setting.
>
> Regarding the baseline comparison, this is not a reporting error. To maintain training parity, we compare with the 400-player set from their appendix (Table 9), which reports results for their 10k-40k+ game unseen players. Although we could not fully reproduce their exact train/test split, our evaluation set is strictly more challenging, since our players generally have less data and we evaluate over a larger candidate pool. For fairness, we train on the same 400-player set they use for their few-shot model and then test on this harder setting. Our method achieves substantially higher accuracy despite the increased difficulty, and we believe this comparison is conservative in their favor rather than ours.

---

### Official Review · Reviewer_MVXi · 2025-10-31

**Soundness:** 2
**Presentation:** 2
**Contribution:** 2
**Rating:** 2
**Confidence:** 5

**Summary:**

This paper addresses the challenge of scalable generative modeling of individual playing styles, proposing a method that combines multi-task learning with parameter-efficient fine-tuning (PEFT) via a Multi-Head Routing (MHR) adapter layer. The key contributions include: (1) a scalable framework for generating individual-specific behaviors without per-user model training, (2) interpretable style vectors enabling style comparison, interpolation, and targeted manipulation, and (3) validation across Chess (47,864 players) and Rocket League (2,000 players), with extensions to image generation. The method achieves high accuracy in player identification, competitive action matching, and efficient style control while significantly reducing computational costs.

**Strengths:**

1. The integration of PEFT (via MHR adapters) with style vectors for scalable individual behavior generation is a creative approach, addressing a critical gap between per-user modeling and scalable systems. I really appreciate this method.
2. Experimental results are good, showing state-of-the-art accuracy in behavioral stylometry (99.8% in Chess, 86.7% in Rocket League) and competitive action matching while reducing computational costs by nearly two orders of magnitude. The cross-domain extension to image generation (CelebA) also highlights generality.

**Weaknesses:**

1. The work focuses heavily on identification and style editing but fails to demonstrate impact on mainstream tasks like skill improvement (e.g., whether style-adapted AIs help humans play better) or more useful abilities. (I want to see how your work has a direct impact on Game-play or other fields)
2. The choice of environments (Chess, Rocket League) is narrow, with Rocket League being less mainstream in AI research compared to platforms like StarCraft or Dota. Testing on more diverse or widely-studied environments would strengthen generalizability claims.
3. The paper does not explicitly analyze how model accuracy evolves as the number of modeled individuals increases. It is also unclear whether adding more users degrades core gaming capabilities.
4. This work has not been compared with SOTA open-sourced models like Llama, Qwen.

**Questions:**

1. Could you clarify the practical impact of your work on mainstream tasks beyond identification and style editing? For example, how could style-adapted AIs facilitate human skill improvement in gaming or deliver value in other domains?
2. Why were only Chess and Rocket League chosen for evaluation? How would your method perform in more mainstream or diverse environments (e.g., StarCraft, Dota) that are widely studied in AI research? Please elaborate on the generalizability across different game genres or non-gaming domains.
3. As the number of modeled individuals scales, how does the model’s identification accuracy evolve? Does adding more users improve, degrade, or leave unchanged the model’s intrinsic gaming capabilities?
4. Have you conducted comparisons with state-of-the-art open-sourced models (e.g., Llama, Qwen) in relevant tasks (e.g., behavioral modeling, style control)? If not, please explain why.

---

> ### Author Response · Authors · 2025-12-03
>
> Thank you for your review. Regarding impact beyond identification and style editing, we believe that effective coaching requires matching the model’s skill to the user’s skill level. When the model is too strong, its suggestions often become unrealistic for lower-skill players, so adapting the model’s behavior is important for producing actionable guidance.
>
> We selected Rocket League and Chess because they offer large, publicly available datasets that can be reliably partitioned by skill. These games differ substantially, and seeing consistent results across both gives us confidence that our method should generalize to other domains. Additional discussion is provided in the paper.
>
> As for comparisons to open-source state-of-the-art models, most do not support action generation in these environments (especially in Rocket League, where actions must be made in real-time), making direct comparison difficult. There is limited similar prior work suitable as a baseline; developing one would require substantial additional effort.

---

### Note · Authors · 2025-12-03

**Comment:**

We thank all the reviewers for their time and constructive feedback. We acknowledge the areas that would benefit from further development in the current version of the work. We are withdrawing the submission and aim to address the concerns in a future iteration.

**Withdrawal Confirmation:**

I have read and agree with the venue's withdrawal policy on behalf of myself and my co-authors.